# Deciphering ion transport and ATPase coupling in the intersubunit tunnel of KdpFABC

Jakob M. Silberberg [1,6], Robin A. Corey[2,6], Lisa Hielkema [3,6], Charlott Stock [1,5], Phillip J. Stansfeld [4✉], Cristina Paulino [3✉] & Inga Hänelt [1✉]

KdpFABC, a high-affinity $K^+$ pump, combines the ion channel KdpA and the P-type ATPase KdpB to secure survival at $K^+$ limitation. Here, we apply a combination of cryo-EM, biochemical assays, and MD simulations to illuminate the mechanisms underlying transport and the coupling to ATP hydrolysis. We show that ions are transported via an intersubunit tunnel through KdpA and KdpB. At the subunit interface, the tunnel is constricted by a phenylalanine, which, by polarized cation-π stacking, controls $K^+$ entry into the canonical substrate binding site (CBS) of KdpB. Within the CBS, ATPase coupling is mediated by the charge distribution between an aspartate and a lysine. Interestingly, individual elements of the ion translocation mechanism of KdpFABC identified here are conserved among a wide variety of P-type ATPases from different families. This leads us to the hypothesis that KdpB might represent an early descendant of a common ancestor of cation pumps.

---

[1] Institute of Biochemistry, Biocenter, Goethe University Frankfurt, Frankfurt/Main, Germany. [2] Department of Biochemistry, University of Oxford, Oxford, UK. [3] Department of Structural Biology, Groningen Biomolecular Sciences and Biotechnology Institute, University of Groningen, Groningen, The Netherlands. [4] School of Life Sciences & Department of Chemistry, University of Warwick, Coventry, UK. [5] Present address: DANDRITE, Nordic EMBL Partnership for Molecular Medicine, Department of Molecular Biology and Genetics, Aarhus University, Aarhus C, Denmark. [6] These authors contributed equally: Jakob M Silberberg, Robin A Corey, Lisa Hielkema. ✉email: phillip.stansfeld@warwick.ac.uk; c.paulino@rug.nl; haenelt@biochem.uni-frankfurt.de

KdpFABC (K[+]-dependent P-type ATPase) is a K[+] pump that is responsible for high-affinity electrogenic K[+] uptake into prokaryotes at very low extracellular K[+] concentrations[1–3]. The heterotetrameric complex combines a P-type ATPase (KdpB) with a channel-like subunit (KdpA) from the superfamily of K[+] transporters (SKT)[4,5]. These subunits are augmented by KdpF, a single transmembrane (TM) helix which stabilizes the complex[6], and KdpC, which has been suggested to influence substrate affinity[7], although its exact function remains elusive. Its chimeric architecture makes KdpFABC unique among K[+] transporters, and ensures the active transport of K[+] with an apparent affinity of 2 μM and with high substrate specificity[2,8–10]. To fulfill this physiological role, both KdpA and KdpB seem to diverge substantially from other members of their families.

The channel-like subunit KdpA consists of four nonidentical, covalently linked membrane-pore-membrane (MPM) motifs, referred to as domains 1–4 (D1–D4), which are arranged in a pseudo-four-fold symmetrical pore. An additional transmembrane helix each is fused at the N and C terminus. KdpA, like other members of the SKT, contains a selectivity filter (SF) that is less conserved than the canonical TVGYG motif of K[+] channels. This has been proposed to reduce the number of K[+] binding sites and suggests a lower ion selectivity[11,12]. However, unlike the SKT members TrkH and KtrB, ion transport by KdpFABC was shown to be highly selective for K[+] over Na[+], and ATPase activation by ammonium or Rb[+] was only possible upon mutation of the KdpA SF region[9,10]. Furthermore, the gating region of the ion channels KtrB and TrkH is not conserved in KdpA. In the latter, the intramembrane loop in D3M2 is much less polar and the highly conserved arginine is not present in D4M2[13].

KdpB is a 7-TM P-type ATPase. Like all other P-type ATPases, KdpB alternates between outward- and inward-open conformations in the so-called Post-Albers cycle, which features states E1 and E2, and the phosphorylated intermediates E1-P and E2-P[14,15]. These states are characterized by large rearrangements of the cytosolic domains (N—nucleotide binding, P—phosphorylation, and A—actuator) of KdpB, which are responsible for nucleotide binding and hydrolysis[12,16,17]. Dephosphorylation is mediated by a TGES motif in the A domain, which is widely conserved among cation P-type ATPases[18–20]. A unique feature of the TGES motif of KdpB is that it is subject to an inhibitory serine phosphorylation (Escherichia coli (Ec)KdpB$_{S162}$). This phosphorylation has been shown to prevent excessive K[+] uptake when external K[+] is high, indicating that it limits uptake by KdpFABC to conditions where it is physiologically necessary[16,21]. Furthermore, KdpB is the smallest known P-type ATPase[22], and was long assumed to be unable to transport substrates itself. Instead, it was suggested to solely provide the energy for ion translocation through the channel-like subunit KdpA[23]. However, the spatial separation of the subunits purported to be responsible for ATP hydrolysis and ion transport posed a conundrum as to how the two processes are coupled.

Several recent structural studies of EcKdpFABC have provided possible explanations of how the unique structural features of both KdpA and KdpB effect K[+] transport. The first X-ray structure of KdpFABC solved in an E1 state led to the identification of an aqueous tunnel within the complex, running along the membrane plane and extending from the SF in KdpA to the P-type ATPase canonical substrate binding site (CBS) in KdpB[16]. Initially, the coupling helix model was proposed, in which the presence of K[+] ions in the SF of KdpA is communicated to KdpB via a Grotthuss (also known as a proton wire) mechanism running through the intersubunit tunnel[16]. KdpB hydrolyzes ATP, undergoes a conformational change, and pulls open the intramembrane gate in the channel subunit via a coupling helix,

thereby enabling K[+] flux through KdpA. The periplasmically oriented KdpC could function as external gate.

For this model to work, a salt bridge between KdpB and KdpA is required to pull on the coupling helix. This salt bridge has, however, been shown to be dispensable, as it was broken in a subsequent cryo-EM structure of KdpFABC in an E2 state[17]. In this structure, virtually no conformational changes in KdpA and KdpC were observed when compared to the E1 state. Instead, the intersubunit tunnel was restricted at the KdpA–KdpB interface, while a new half-channel in KdpB from the CBS to the cytosol opened. This is accomplished by a significant reorientation of the positively charged KdpB$_{K586}$ in the CBS and the surrounding TM helices. Finally, a cryo-EM map of KdpFABC in an E1 state contained several nonprotein densities within the intersubunit tunnel, which were suggested to be K[+] ions.

Based on these structural insights, the intersubunit tunnel translocation model was suggested[17]: in the E1 state, extracellular K[+] ions are forwarded from the SF in KdpA through the intersubunit tunnel to the CBS in KdpB, where they trigger ATP hydrolysis and phosphorylation of the catalytic KdpB$_{D307}$. The ensuing E1-P/E2-P transition is associated with a reorientation of KdpB$_{K586}$, which pushes K[+] out of the CBS and simultaneously acts as a built-in counterion to stabilize the E2/E1 transition. Recent cryo-EM structures extend the model by suggesting a displacement of the ion from the CBS into an adjacent low-affinity site in the E2-P state, from where ion release into the cytoplasm is favored[12].

Here, we decipher the molecular basis of the transport pathway of KdpFABC by addressing how K[+] ions are forwarded through the tunnel, how they bind to the CBS, and the mechanisms by which ATP hydrolysis and the subsequent E1-P/E2-P transition are triggered. We present two cryo-EM structures of KdpFABC in an E1·ATP state, loaded with K[+] and Rb[+] ions, respectively, allowing a more unambiguous assignment of ion positions in the complex. Further, we characterize the dynamics of ion coordination and translocation through the intersubunit tunnel by molecular dynamics (MD) simulations. Finally, we combine biochemical and MD approaches to identify three conserved residues responsible for ATPase coupling, ion propagation through the intersubunit tunnel into the CBS, and regulation of the rate-limiting step of turnover, providing the foundation for K[+] transport by KdpFABC.

## Results

**Structures of K[+]- and Rb[+]-loaded KdpFABC in an E1·ATP state.** In various cryo-EM structures of KdpFABC, nonprotein densities have been observed along the intersubunit tunnel. Although the densities could not be unambiguously assigned, they were suggested to represent K[+] ions or water molecules[12,16,17]. Due to its higher atomic number, Rb[+] ($Z = 37$) scatters electrons more strongly than K[+] ($Z = 19$) by an expected factor of 2.5[24–26]. We set out to exploit this as a strategy to assign substrate ions with more confidence. To this end, we determined cryo-EM structures of the complex loaded with K[+] and Rb[+], both arrested in an E1·ATP state. In the K[+]-loaded sample, we used the mutation KdpB$_{D307N}$, which precludes ATP hydrolysis, to stall the complex in the E1 state. To incorporate Rb[+], we employed variant KdpFA$_{G232D}$B$_{S162A}$C, in which a central glycine (KdpA$_{G232}$) of the KdpA SF is mutated to allow passage of Rb[+][9]. K[+] ions co-purified in the intersubunit tunnel of the variant were exchanged for Rb[+] by applying turnover conditions. Here, the KdpB$_{S162A}$ mutation prevented the inhibitory phosphorylation in the A domain, enabling turnover.

In both samples, the complex was trapped by the addition of the nonhydrolysable ATP analog AMPPCP prior to cryo-EM

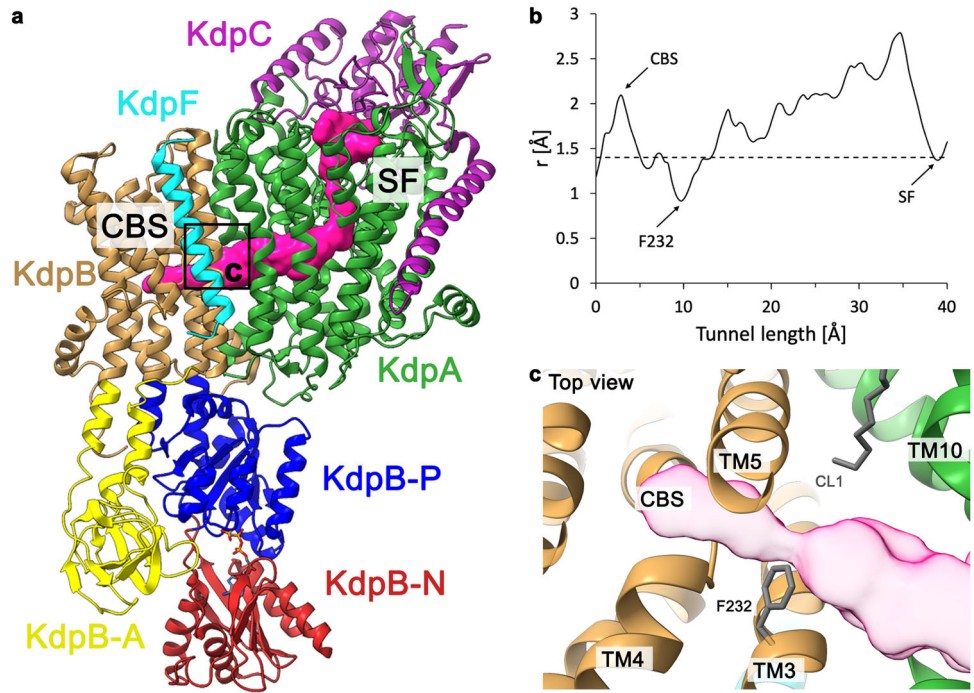

**Fig. 1 Intersubunit tunnel in the K⁺-loaded E1·ATP structure of KdpFABC. a** Structural model of E1·ATP KdpFAB$_{D307N}$C in ribbon representation. The following color code is retained for all following figures unless otherwise specified: KdpA: green, KdpC: purple, KdpF: cyan, KdpB: TMD sand, N domain: red, P domain: blue, and A domain: yellow. Here, the KdpA pore entrance at the SF and intersubunit tunnel leading to the CBS are visualized as a pink surface. **b** Radius of the intersubunit tunnel along its length, from below the KdpA SF to the KdpB CBS. The tunnel is wide enough to accommodate K⁺ (r = 1.4 Å, dashed line) along nearly its entire length, with a significant constriction immediately before the CBS. **c** Top view of the constriction in the intersubunit tunnel, which is mediated mainly by KdpB$_{F232}$ as it protrudes into the tunnel.

sample preparation. The structures were determined at a global resolution of 3.1 and 3.2 Å in the K⁺- and Rb⁺-loaded samples, respectively, with higher local resolution in the transmembrane region. Both structures are nearly identical and adopt an E1·ATP conformation (Fig. 1a, Supplementary Figs. 1–6, Supplementary Table 1). The nucleotide AMPPCP is coordinated between the N and P domains, reflecting the common catalytic ATP binding mode of P-type ATPases[27] (Supplementary Fig. 6b, c). This state shows an outward-open half-channel that reaches from the KdpA SF to the CBS in KdpB (Fig. 1a). The tunnel is sufficiently wide to accommodate either K⁺ or Rb⁺ ions along its entire length. There is a constriction point at the SF in KdpA, where K⁺ is dehydrated, and a second constriction immediately before the CBS (Fig. 1b, Supplementary Fig. 6d). This constriction is mainly formed by a single residue, namely KdpB$_{F232}$ (Fig. 1c).

**Cardiolipin aids the E1-P/E2-P transition in KdpFABC.** In both structures, we identify two cardiolipin (CL) molecules bound to KdpFABC (Fig. 2a, Supplementary Fig. 6e, Supplementary Fig. 7a, b). The assignment as cardiolipin in these positions is strongly supported by coarse-grained MD simulations showing preferred CL binding (Fig. 2b, Supplementary Fig. 7c). CL binding at the KdpA–KdpB interface (CL1) is principally coordinated by two basic residues (KdpA$_{R278}$ and KdpB$_{R651}$), a glycine residue, a polar residue and an aromatic residue (KdpA$_{G524}$, KdpA$_{H523}$, KdpA$_{W285}$) making it a typical high-affinity *E. coli* CL binding site[28]. CL1 appears to be of integral structural importance, as evidenced by the loss of KdpFABC in vivo when the binding site is mutated (Fig. 2c, Supplementary Fig. 7e). One hydrophobic chain of CL1 reaches towards the intersubunit tunnel, lying immediately opposite KdpB$_{F232}$ (Supplementary Fig. 7d). Since CL specifically stimulates the ATPase activity of

KdpFABC, it may support the transition to the E2 conformation, as previously proposed[12] (Fig. 2d).

**Ions are transported through the intersubunit tunnel.** Both cryo-EM structures feature multiple distinct nonprotein densities filling the entire length of the outward-open half-channel, from the SF in KdpA to the CBS in KdpB (Fig. 3a). The individual positions in both structures closely overlap (Fig. 3a, Supplementary Fig. 9a). Compared to our previous E1 cryo-EM structure 6HRA, a higher number of densities is observed within the K⁺-loaded tunnel, which we attribute to the increased K⁺ concentration used for sample preparation (50 mM vs. 1 mM) and the improved resolution of the transmembrane domain[17] (Supplementary Fig. 8a). Potential coordinating residues were identified for the majority of these positions (Supplementary Figs. 8b, 9b, Supplementary Table 2), and the tunnel is wide enough to accommodate additional waters for solvation. Together, these factors already support the assignment of the additional densities as ions.

Notably, the K⁺-loaded structure features densities in the S1, S3, and S4 positions of the SF of KdpA, although the S4 site was previously suggested to be lost in KdpFABC[12] (Supplementary Fig. 8c). The nonprotein density in the S1 site is absent in the Rb⁺-loaded structure, probably because the mutation KdpA$_{G232D}$ changes the SF geometry (Supplementary Fig. 9c). The loss of a coordination site in the SF might also explain the reduced affinity of KdpFA$_{G232D}$BC for K⁺ compared to wild-type KdpFABC[10].

When comparing the individual nonprotein densities in both structures low-pass filtered to the same resolution, the densities in the KdpA SF S3 site and the KdpB CBS are increased *ca.* two-fold in intensity in the presence of Rb⁺ (Fig. 3b–d), close to the expected factor of 2.5[24–26]. This suggests that the densities at both the beginning and the end of the intersubunit tunnel indeed correspond to ions, not waters, and, consequently, that the

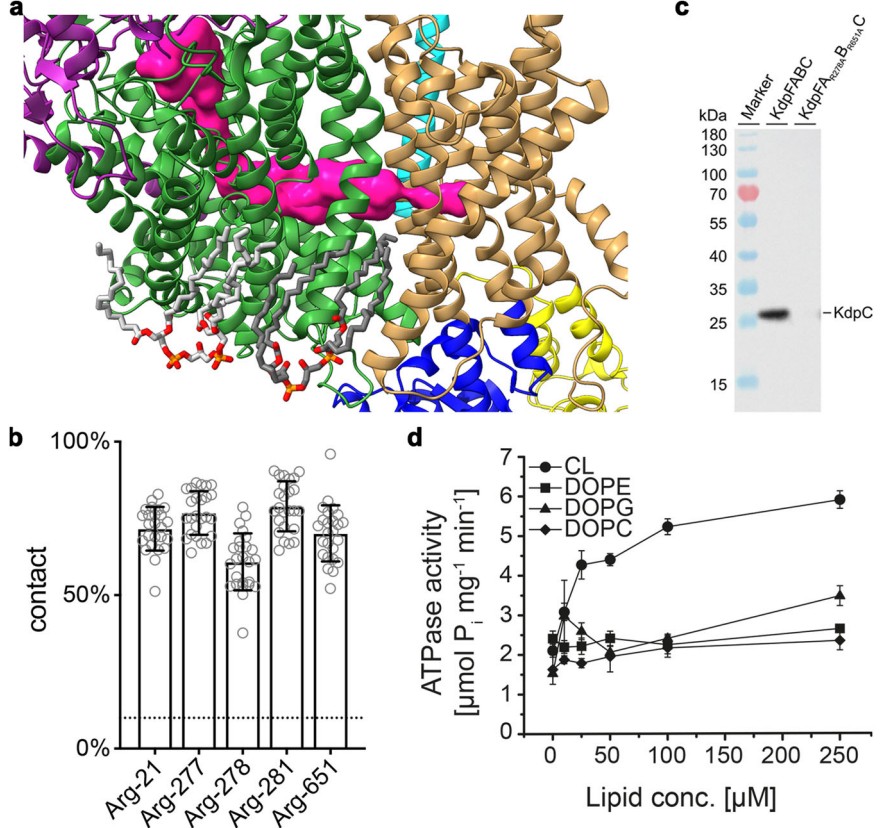

**Fig. 2 Cardiolipin in KdpFABC. a** Two identified cardiolipin (CL) molecules (gray) bound to KdpFABC. A hydrocarbon tail of CL1 (dark gray) extends into the complex at the KdpA/KdpB interface. **b** Pooled CL contact probabilities of selected residues, from five representative structures in various E1 and E2 conformations (5MRW, 6HRA, 6HRB, 7BGY, and 7NNL), measured as proportion of frames in a coarse-grained MD simulation the CL and the respective residue are in contact, i.e., within a 0.6 nm cutoff distance. The concentration of CL in the membrane (10%) is shown as a dotted line, indicating a very high accumulation of CL interactions at these residues. Data points represent the average and error bars the standard deviation from 25 independent simulations (5 each per structure). **c** Anti-His Western Blot of whole-cell samples of *E. coli* LB2003 cells carrying KdpFABC variants, detecting His-tagged KdpC. The simultaneous knockout of residues coordinating CL1 in KdpFA$_{R278A}$B$_{R651a}$C abolishes protein production, showing the structural importance of CL binding for complex stability. This result was reproducible in a biological duplicate. **d** CL-dependent ATPase activity of KdpFABC. Rising CL conc. (0–250 μM) stimulate ATPase activity up to two fold, while additions of PE, PG, and PC do not change the ATPase activity of KdpFABC. Increased ATPase activity indicates an acceleration of the rate-limiting E1/E2 transition. Data points represent the average and error bars the standard deviation from a technical triplicate.

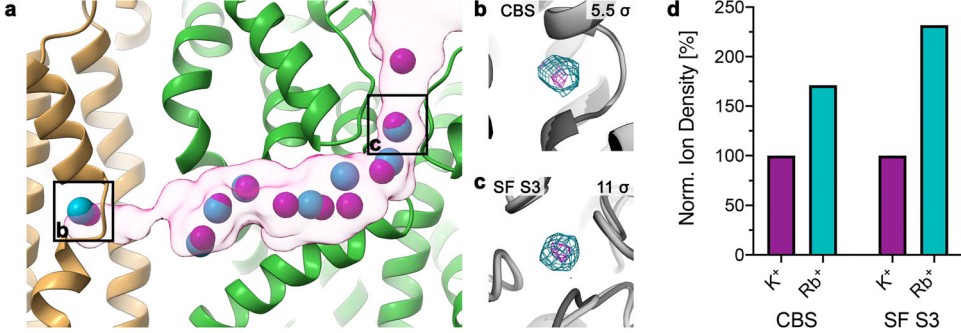

**Fig. 3 Ions traverse the intersubunit tunnel. a** Potential ions in K$^+$- and Rb$^+$-loaded E1·ATP KdpFABC within the intersubunit tunnel, from the KdpA SF (right) to the KdpB CBS (left). Based on the high ion concentrations in each sample, densities were assigned as K$^+$ and Rb$^+$ ions, respectively. K$^+$ visualized in purple, Rb$^+$ in turquoise. **b, c** Comparison of densities in the CBS and in SF position S3, respectively, between the K$^+$- and Rb$^+$-loaded maps when low-pass filtered to 3.2 Å and displayed at the same contour level (σ). Densities in the Rb$^+$-loaded structure (turquoise) are stronger than in the K$^+$-loaded structure (purple), suggesting that they correspond to ions and that Rb$^+$ is integrated into the complex. Contour levels are indicated in the top right corner of each panel. **d** Quantification of the effect of Rb$^+$ substitution in the CBS and SF S3. Densities were low-pass filtered to 3.2 Å, normalized to the K$^+$-loaded structure for each position, and indicate a *ca.* two-fold signal increase in the Rb$^+$-loaded structure in both positions. For the full, non-normalized comparison of all ion densities, see Supplementary Fig. 9d.

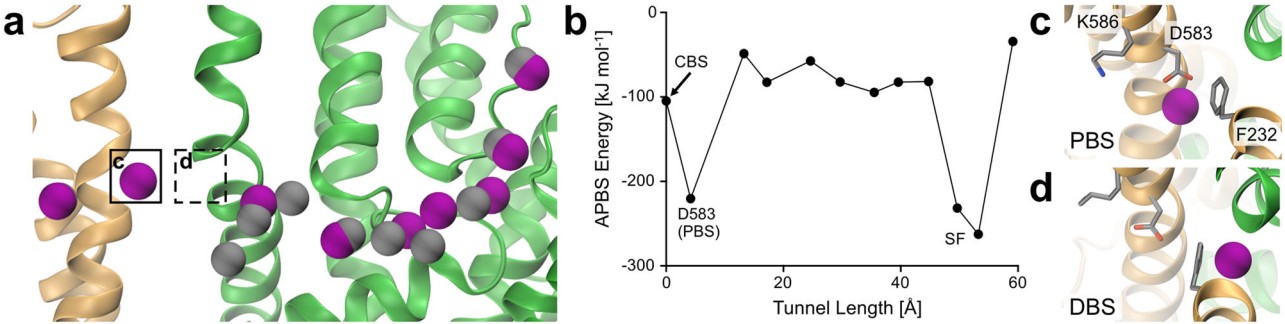

**Fig. 4 Occupancy and energetics of K⁺ ions in the intersubunit tunnel in MD simulations. a** Comparison of K⁺ positions from cryo-EM (gray) and at the endpoint of a short atomistic MD relaxation (purple). The ions rearrange slightly but remain in the intersubunit tunnel, indicating its viability for translocation. A new ion position is taken up adjacent to the CBS. **b** APBS coordination energy profile for K⁺ along the length of the intersubunit tunnel. Coordination is optimal at the KdpA SF, where ions are fully dehydrated by the protein, and immediately before the KdpB CBS, where they are bound by KdpB$_{D583}$. **c, d** New coordination sites around KdpB$_{F232}$ observed in atomistic MD simulations. During the progression towards the CBS, ions are likely coordinated by cation–π interactions on both faces of KdpB$_{F232}$. Coordination in the PBS (**c**) is strong, with coordination supported by KdpB$_{D583}$. Coordination in the DBS (**d**) is transient.

tunnel constitutes the ion translocation pathway. The effect is not as evident for ions in the middle of the intersubunit tunnel (Supplementary Fig. 9d), probably due to more dynamic and nonspecific ion coordination at these sites compared to the KdpA SF and KdpB CBS. Alternatively, these densities could represent waters in the intersubunit tunnel, as proposed recently[12], although waters are not typically seen in EM at the obtained resolutions[29]. An unequivocal assignment of ions would require anomalous dispersion data from an X-ray diffraction approach.

The viability of ion translocation through the intersubunit tunnel was further probed by atomistic MD simulations of K⁺-loaded KdpFABC. The data revealed that the ions rearranged slightly during a short 20 ns MD relaxation, but are otherwise well coordinated in the tunnel (Fig. 4a). The simulations also showed that, in addition to direct protein contacts, water molecules form solvation shells around the K⁺ ions in the tunnel (Supplementary Fig. 10a). An analysis of the coordination energy for K⁺ along the tunnel by Adaptive Poisson-Boltzmann Solver (APBS) analysis was used as an approximate framework to compare ion positions in the tunnel, and indicated a favorable energy profile along the entire length, with notable energy minima in the KdpA SF, where K⁺ is fully dehydrated by the protein, and, interestingly, at the end of the intersubunit tunnel, just before the KdpB CBS (Fig. 4b, Supplementary Fig. 10b, c). Notably, this new ion position was taken up immediately, where K⁺ is primarily coordinated by KdpB$_{D583}$. In addition, KdpB$_{F232}$ aids ion coordination, either by direct cation–π stacking or by CH–π interaction with KdpB$_{L262}$, which in turn coordinates the ion with its backbone carbonyl (Fig. 4c, Supplementary Fig. 10d). Since this position is located on the proximal side of KdpB$_{F232}$ with respect to the CBS, we termed it the proximal binding site (PBS). When the five ions closest to the CBS were deleted, simulations revealed a rapid progression of two ions from KdpA towards the CBS (Video 1). The first ion moves quickly past the constriction formed by KdpB$_{F232}$ to the described energy minimum, coordinated by KdpB$_{D583}$ and the proximal phenylalanine face (PBS, Fig. 4c). Subsequently, the second ion moves along the intersubunit tunnel in a similar fashion, until being coordinated by cation–π stacking at the face of KdpB$_{F232}$ distal to the CBS (distal binding site, DBS, Fig. 4d). These simulations not only support the translocation of ions through the intersubunit tunnel, but also suggest that ion propagation is supported by a pull-on mechanism mediated by the KdpB$_{D583}$ energy well in the PBS at the end of the tunnel.

Moreover, the ion positions taken up in the MD simulations suggest that cation–π interactions with KdpB$_{F232}$ play an important role in the progression of ions towards the CBS. Since the coordination in the PBS is energetically similar to that observed in the KdpA SF, an external force is likely required to push the ion onward into the CBS for transport, as is the case in the SF. The MD simulations used a point-charge force field, thus this process could not be fully resolved[30]. However, we postulate that the second ion found in the MD simulations, coordinated at the DBS, repels the first ion from the PBS, pushing it forward towards the CBS for subsequent transport. This could occur either through perturbation of the polarity of the phenylalanine π-electron clouds or simply through charge–charge repulsion, comparable to the proposed Coulomb knockon in the SFs of K⁺ channels[31,32].

**Three residues mediate ion progression and ATPase coupling.** To further elucidate the proposed role of KdpB$_{F232}$ in ion translocation, we mutated the residue to an isoleucine, which has a bulky side chain that fills the tunnel similarly to KdpB$_{F232}$, but is unable to coordinate K⁺ via cation–π stacking. The KdpFAB$_{F232I}$C variant showed a similar K⁺-coupled ATPase activity as the wild type, but with an ~80% reduced transport rate (Fig. 5a–c). Atomistic MD simulations showed that the mutation does not greatly affect ion progression towards KdpB, attracted by the KdpB$_{D583}$ energy well, and PBS and CBS can be populated. However, the DBS adjacent to KdpB$_{F232I}$ is lost, and ion binding at this site is no longer observed (Fig. 5d). The loss of ion coordination, together with the reduced transport rate, supports our hypothesis that the DBS next to KdpB$_{F232}$ is required for the efficient forwarding of a K⁺ ion from the PBS to the CBS.

The fact that the ATPase activity is undisturbed in KdpFAB$_{F232I}$C further suggests that cation binding to the PBS is sufficient to trigger ATP hydrolysis. We postulate that the neutralization of the negative charge of KdpB$_{D583}$ by cation binding could be the stimulus, as variant KdpFAB$_{D583A}$C has previously been shown to uncouple ATP hydrolysis from K⁺ and abolish K⁺ transport[33]. The lack of transport is likely based on the loss of the energy well at the end of the intersubunit tunnel, as MD simulations with KdpFAB$_{D583A}$C showed that ions are no longer effectively pulled past the constriction formed by KdpB$_{F232}$ (Fig. 5d). To elucidate whether neutralization at the PBS is in fact responsible for stimulating ATP hydrolysis, we

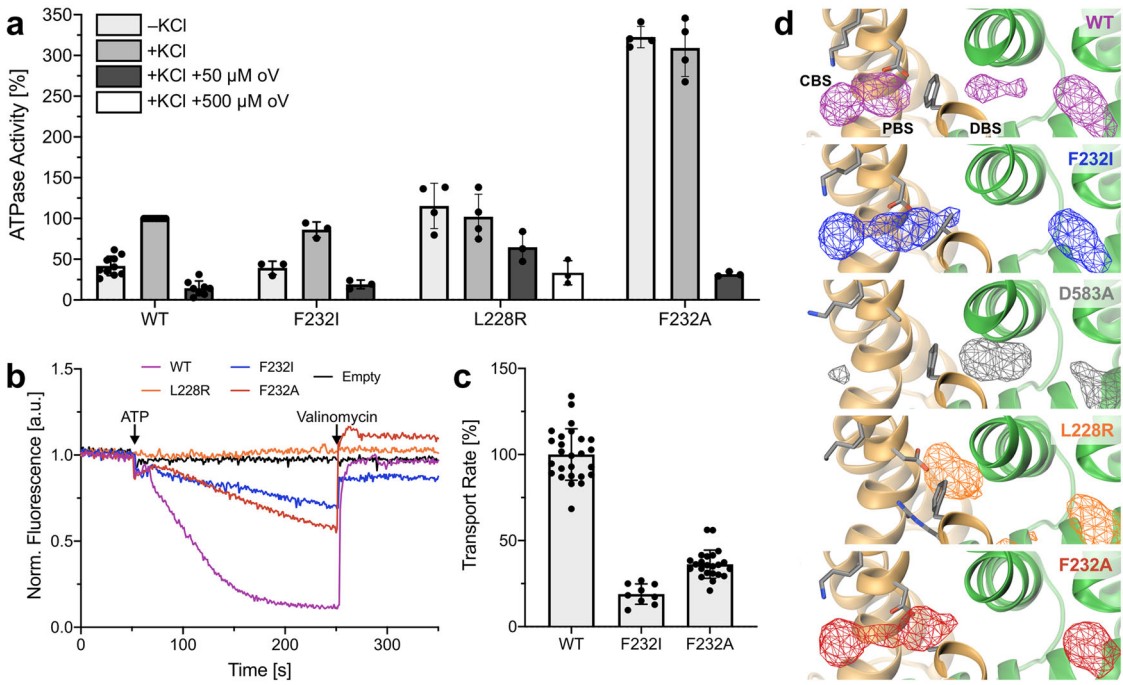

**Fig. 5 Activity and K⁺ occupancy in KdpFABC variants with mutations around the CBS. a** ATPase activity of KdpFABC variants in the absence and presence of K⁺ and the P-type ATPase inhibitor orthovanadate (oV), normalized to K⁺-stimulated activity of wild-type (WT) KdpFABC. KdpFAB$_{F232I}$C retains K⁺-stimulated ATPase activity. The K⁺ mimic KdpB$_{L228R}$ uncouples ATP hydrolysis from K⁺, and increases resistance to orthovanadate. KdpB$_{F232A}$ uncouples ATP hydrolysis from K⁺ and largely increases the activity, while maintaining orthovanadate inhibition. Data points represent the average and error bars the standard deviation of measurements from technical replicates (–KCl | +KCl | +KCl +50 µM oV | +KCl +500 µM oV: WT $n = 10 | 11 | 9 | 0$, F232I $n = 3 | 3 | 3 | 0$, L228R $n = 4 | 4 | 3 | 3$, F232A $n = 4 | 4 | 3 | 0$) from at minimum biological triplicates (WT $n = 9$, F232I $n = 3$, L228R $n = 3$, F232A $n = 3$). **b, c** Transport activity of liposome-reconstituted KdpFABC variants. Decreasing ACMA fluorescence indicates K⁺ transport. Initial slopes were fitted linearly to obtain transport rates. Contrasting their ATPase activity, transport by KdpFAB$_{F232I}$C and KdpFAB$_{F232A}$C is slower than that of WT (reduced by 80 and 65%, respectively). KdpFAB$_{L228R}$C showed no transport. Data points represent the average and error bars the standard deviation of measurements from technical replicates (WT $n = 26$, F232I $n = 9$, F232A $n = 23$) from at minimum biological triplicates (WT $n = 8$, F232I $n = 3$, F232A $n = 3$). **d** Computed densities of K⁺ (mesh) in atomistic MD simulations of KdpFABC variants. In WT, ions can occupy three main sites: the DBS and PBS at the KdpA/KdpB interface before and behind the constricting residue KdpB$_{F232}$, and the CBS in KdpB. In KdpFAB$_{F232I}$C, ions can reach the PBS, coordinated by KdpB$_{D583}$, but the DBS is lost. No significant ion passage past KdpB$_{F232}$ was observed in KdpFAB$_{D583A}$C and KdpFAB$_{L228R}$C. Removing the steric hindrance in KdpFAB$_{F232A}$C once again allows ion passage to the PBS, but the DBS remains lost. Ion progression into the CBS in these simulations is possible because KdpB$_{K586}$ is set as uncharged.

placed a positively charged side chain into the PBS, mimicking K⁺ binding. The activity of KdpFAB$_{L228R}$C fully mirrored that of KdpFAB$_{D583A}$C: the ATPase activity level was similar to WT and uncoupled from K⁺, K⁺ transport was abolished, and in MD simulations no ions passed KdpB$_{F232}$ (Fig. 5a, b, d).

In addition to its role in ion coordination at the PBS and forwarding of K⁺ to the CBS, KdpB$_{F232}$ appears to present a significant steric constriction in the intersubunit tunnel, preventing ion transition to the PBS unless pulled by the negatively charged KdpB$_{D583}$ (Fig. 5d). To probe this, we investigated the impact of the KdpB$_{F232A}$ mutation where, unlike KdpB$_{F232I}$, not only the possibility of cation–π interactions, but also steric hindrance is abolished. As seen for KdpB$_{F232I}$, ions can reach the CBS and PBS but do not occupy the DBS in the MD simulations (Fig. 5d). Moreover, simulations on the double mutant KdpB$_{F232A/D583A}$ showed that ions can reach the CBS even in the absence of the KdpB$_{D583A}$ energy well when the steric hindrance is removed (Supplementary Fig. 10e). Functional assays of KdpFAB$_{F232A}$C, like KdpFAB$_{F232I}$C, revealed a significantly reduced transport rate (Fig. 5b, c), supporting the importance of cation-π interactions and the DBS for efficient ion progression towards the CBS. Unexpectedly however, KdpFAB$_{F232A}$C showed a K⁺-uncoupled and enhanced ATPase activity (Fig. 5a). The uncoupling suggests that ATP hydrolysis is stimulated by a K⁺-independent

neutralization of the PBS, as described above for KdpFAB$_{D583A}$C and KdpFAB$_{L228R}$C. A possible explanation could be an unspecific protonation of KdpB$_{D583}$ via a continuous water wire along the intersubunit tunnel, which is otherwise sterically blocked in the WT and KdpFAB$_{F232I}$C. Thus, in addition to being an element in ion coordination and progression, KdpB$_{F232}$ may also fulfill a gatekeeper role in KdpFABC by preventing unspecific stimulation of ATP hydrolysis by K⁺-independent neutralization of KdpB$_{D583}$.

**KdpB$_{F232}$ modulates the rate-limiting step in KdpFABC.** In addition to uncoupling ATP hydrolysis from K⁺, KdpFAB$_{F232A}$C also shows a three- to four-fold increase in $V_{max}$ of ATP hydrolysis when compared to stimulated wild-type KdpFABC (Fig. 5a, Supplementary Fig. 11). By contrast, the $K_m$ value for ATP is similar. Interestingly, the uncoupled variants KdpFAB$_{L228R}$C and KdpFAB$_{D583A}$C[33] showed a $V_{max}$ comparable to that of stimulated wild-type KdpFABC. This discrepancy between the uncoupled variants suggests that ATPase uncoupling and deregulation are separate processes. The KdpFAB$_{D583A}$C variant was previously shown to have a decreased sensitivity to orthovanadate, which was interpreted as indicative of an accelerated E2/E1 transition, as orthovanadate binds to the protein in the post-hydrolysis E2 state and stalls turnover[33]. Similarly, the

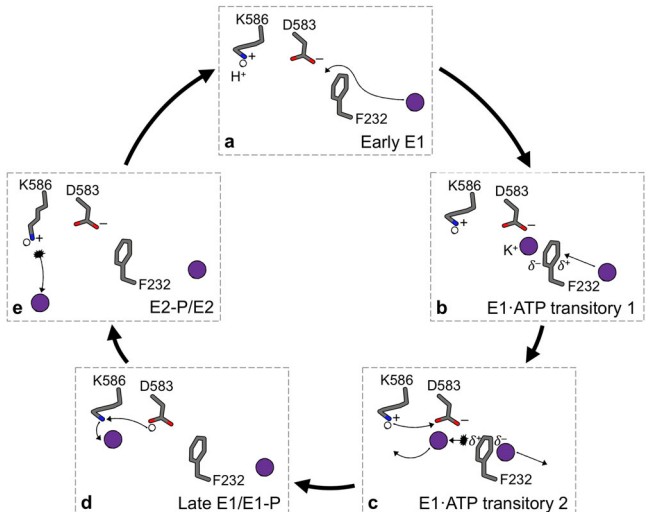

**Fig. 6 Molecular mechanism of ion progression in the CBS during the KdpFABC transport cycle. a** In the E1 ground state (Early E1), KdpB$_{K586}$ is protonated and KdpB$_{D583}$ deprotonated, forming the energy well at the end of the intersubunit tunnel. **b** ATP binds and a first ion is rapidly pulled past the constriction of KdpB$_{F232}$ into the PBS, coordinated by KdpB$_{D583}$ and assisted by cation–π stacking with KdpB$_{F232}$ (E1·ATP transitory 1). **c** When a second ion reaches the DBS, π cloud polarization is perturbed, slightly repelling the first ion forward towards the CBS (E1·ATP transitory 2). **d** Upon a protonation switch between KdpB$_{D583/K586}$, the PBS ion moves forward into the CBS, with coordination supported by KdpB$_{K586}$ (Late E1/E1-P). The energy well in the PBS is lost by protonation of KdpB$_{D583}$, preventing the ion from being pulled back or a second ion from being pulled forward from the DBS. **e** Upon E1-P/E2-P transition, the protonation switches back to its ground state, with a conformational shift of the now positively charged KdpB$_{K586}$ forcing the ion out of the CBS (**5**—E2-P/E2). In the E2/E1 transition, KdpB$_{K586}$ acts as a built-in counterion.

KdpB$_{L228R}$ mutation led to an increased resistance to inhibition by orthovanadate (Fig. 5a). The proposed accelerated E2/E1 transition could be explained by the lack of ions in the CBS in these variants, which otherwise have to be displaced before the return to the E1 state. By contrast, KdpB$_{F232A}$ does not increase the resistance to orthovanadate, suggesting that the mutation must affect the catalytic cycle differently. We propose that the increase in ATPase activity observed for KdpFAB$_{F232A}$C is based on the acceleration of the rate-limiting step in KdpFABC turnover, the E1-P/E2-P transition[34], which is characterized by large rearrangements of the A domain[12,17]. KdpB$_{F232}$ lies in TM3, which is directly connected to the A domain (Supplementary Fig. 12a). Therefore, the plasticity of the A domain could be modulated by the large phenylalanine side chain, regulating the E1-P/E2-P transition. Thus, in addition to its involvement in ion coordination and gatekeeping the CBS, KdpB$_{F232}$ may modulate the movement of the KdpB-A domain, in part regulating the rate-limiting step of KdpFABC turnover.

## Discussion

By combining structural, biochemical, and MD simulation data, we demonstrate how K$^+$ is translocated through KdpFABC. Beyond confirming the previously proposed intersubunit tunnel translocation model, our data allow us to propose a molecular mechanism for ion propagation and ATPase coupling, particularly involving residues KdpB$_{F232}$, KdpB$_{D583}$, and KdpB$_{K586}$ (Fig. 6). Notably, the here proposed model also explains previously reported functional effects of mutations in and around the CBS (for details see Supplementary Fig. 13).

In the E1 state, a first K$^+$ ion is drawn past the constriction formed by KdpB$_{F232}$ to the energy minimum at the PBS, generated largely by the negatively charged KdpB$_{D583}$ (Fig. 6a). Coordination is aided in part by direct cation–π stacking to KdpB$_{F232}$, or by KdpB$_{L262}$, which in turn is CH–π stacked to KdpB$_{F232}$. The arrival of a second ion from the tunnel at the DBS perturbs the polarization of the KdpB$_{F232}$ π-system and electrostatically repels the ion in the PBS towards the CBS (Fig. 6b). We propose that this induces a protonation switch between KdpB$_{D583}$ and KdpB$_{K586}$, neutralizing both (Fig. 6c). This hypothesis is supported by their predicted pKa values close to physiological pH (pKa of KdpB$_{K586}$: 7.5; pKa of KdpB$_{D583}$: 7.2), suggesting that protonation swapping is energetically inexpensive. The proton switching consequently allows the ion in the PBS to move forward into the CBS without being pulled back by the energy well (Fig. 6d). Coordination in the CBS is facilitated by deprotonated KdpB$_{K586}$, as observed in previous E1 structures of KdpFABC[12,16]. This forwarding of a K$^+$ ion by a protonation switch mechanism is in good agreement with ion behavior in MD simulations: in the doubly charged state, the ion binds strongly at the PBS; when the protonation states are switched (doubly neutral state), the ion is repelled forward into the CBS once a second ion approaches the DBS (Supplementary Fig. 14a, b).

During the post-hydrolysis E1-P/E2-P transition, the protonation states of KdpB$_{D583/K586}$ likely revert back to their initial states, such that the now positively charged lysine side chain displaces the K$^+$ from the CBS, as previously proposed[17] (Fig. 6e). A lipid at the KdpA/KdpB interface might support the E1-P/E2-P transition, as it was suggested to stabilize the E2 conformation[12]. We assign the lipid as CL (Fig. 2, Supplementary Fig. 7), and the accelerated turnover in the presence of CL probably reflects the effect on the rate-limiting step. Similar lipid-mediated effects have been shown for other P-type ATPases[35,36].

Recent structural observations in the E2 state suggest that KdpB$_{K586}$ pushes the ion into an adjacent low-affinity site, from where it is released through a previously observed inward-open half-channel into the cytosol[12,17]. We tested this hypothesis using MD simulations, and saw that ions were released from the CBS in the E2 state via the proposed half-channel, but did not enter this low-affinity release site (Supplementary Fig. 14c). Moreover, ions placed in this site rapidly moved back towards the CBS, as shown by K$^+$ distances to KdpB$_{K586}$ in the CBS (Supplementary Fig. 14d). From there, they followed the same exit pathway as before, indicating that the proposed low-affinity site likely does not represent a K$^+$ binding site. In accordance with these observations, the physiological necessity for such a low-affinity site remains unclear, as ion release appears to be possible without traversing it.

The duration of ion release from the CBS is variable, taking up to 250 ns in simulations (Supplementary Fig. 14e). This may explain the accelerated E2/E1 transition in KdpFAB$_{D583A}$C and KdpFAB$_{L228R}$C, in which ions do not reach the CBS and no longer need to be displaced. During the E2/E1 transition, KdpB$_{K586}$ acts as a built-in counterion, as previously suggested[17] (Fig. 6e). Alternating access is granted by a constriction at the KdpA/KdpB interface in the E2 state, which prevents new ions from entering the CBS before the return to the E1 conformation (Supplementary Fig. 14f).

But at what point in the proposed transport cycle does ATP hydrolysis occur? Our data suggest that hydrolysis is induced by the binding of K$^+$ to the PBS, which neutralizes KdpB$_{D583}$ and leaves KdpB$_{K586}$ at the CBS positively charged (Fig. 6b, c). This trigger is possibly mediated through TM5 of KdpB, harboring KdpB$_{D583}$ and KdpB$_{K586}$, which directly connects to the P domain (Supplementary Fig. 12b). Hence, to allow K$^+$ transport to the cytoplasm, the forwarding of the ion from the PBS to the CBS

**Table 1 KdpFABC variants used in this study.**

| Variant | Mutation purpose | Experiment |
|---|---|---|
| KdpFABC | Wild type | ATPase, transport, complementation |
| KdpFAB$_{D307N}$C | Catalytically inactive variant, stalled in E1 state | Cryo-EM, MD, complementation |
| KdpFA$_{G232D}$B$_{S162A}$C | Rb$^+$-permissive mutation in SF, no inhibitory phosphorylation | Cryo-EM |
| KdpFA$_{R278A}$BC | CL1-coordinating residue | Complementation |
| KdpFAB$_{R651A}$C | CL1-coordinating residue | Complementation |
| KdpFA$_{R278A}$B$_{R651A}$C | CL1-coordinating residues | Complementation |
| KdpFAB$_{F232I}$C | Coupling residue in intersubunit tunnel | ATPase, transport, MD |
| KdpFAB$_{D583A}$C | Coupling residue in CBS | MD |
| KdpFAB$_{L228R}$C | K$^+$ mimic in PBS | ATPase, transport, MD |
| KdpFAB$_{F232A}$C | Coupling residue in intersubunit tunnel | ATPase, transport, MD |
| KdpFA$_{Q116R}$BC | Affinity-reducing mutation in SF | MD |
| KdpFAB$_{K586R}$C | Coupling residue in CBS | MD |
| KdpFAB$_{F232A/D583A}$C | Coupling residues in intersubunit tunnel and CBS | MD |

must be faster than the transition from E1·ATP to the inward-open E2-P. This scenario is plausible, as the E1-P/E2-P transition is the rate-limiting step of KdpFABC[34], regulated by residue KdpB$_{F232}$.

In fact, our data place a particular importance on residue KdpB$_{F232}$. It appears to have a three-fold role in KdpFABC turnover: (1) it acts as a gatekeeper to prevent unspecific access to the PBS and CBS from the intersubunit tunnel, thereby preventing an uncoupling of ATP hydrolysis from K$^+$ ; (2) it is involved in ion coordination and progression via its π-electron system; and (3) it regulates the rate-limiting step of KdpFABC, the E1-P/E2-P transition. As such, KdpB$_{F232}$ directly links ion propagation and turnover. The elaborate coupling system may explain the low transport rate of KdpFABC compared to other P-type ATPases and the strict coupling of transport to ATP hydrolysis[37]. Accordingly, KdpB$_{F232}$, like KdpB$_{D583/K586}$[33], is highly conserved among KdpB sequences from different species (Supplementary Fig. 15).

Notably, all elements involved in ion progression and coupling, individually or in combinations, have also been proposed for other P-type ATPases. Cation–π coordination by phenylalanine residues has been implicated in ion transport by the Na$^+$/K$^+$-ATPase and in the sarcoplasmic Ca$^{2+}$ pump SERCA. In the Na$^+$/K$^+$-ATPase (P2C family), a phenylalanine is involved in the dehydration of ions entering the CBS[38]. Like KdpB$_{F232}$, this residue lies in the TM helix connected to the A domain, and its mutation to alanine resulted in an acceleration of the E1-P/E2-P transition[39,40]. In SERCA (P2A family), a structurally equivalent phenylalanine residue was suggested to be involved in stabilizing the E2 conformation, favoring the E1-P/E2-P transition[41]. In the Zn$^{2+}$ ATPase ZntA (P1B family), metal binding and ATPase coupling are facilitated by an aspartate/lysine pair in the CBS, while a conserved tyrosine residue controls access to the CBS[42,43]. The lysine also acts as a built-in counterion in the E2 state, enabling electrogenic transport without a counterion, as in KdpFABC. A similar mechanism was shown for the H$^+$ ATPase (P3A family), in which the CBS is formed by an arginine and an aspartate, while an asparagine acts as an H$^+$ gatekeeper, replacing the phenylalanine/tyrosine in KdpFABC and ZntA, respectively[44,45]. This indicates that metal ATPases require the aromatic residue for interaction with the ion. The H$^+$/K$^+$ ATPase (P2C family) features a lysine in the CBS that is suggested to expel the primary ion in the E2 state before counterion binding[46], in part reflecting the function proposed for KdpB$_{K586}$. Prevention of counterion binding was also observed upon mutation of an arginine into the corresponding position of the Na$^+$/K$^+$ ATPase[47]. The fact that different elements identified in various P-type ATPase families are all present in KdpB suggests that

it may be a close descendant of a common ancestor of cation pumps.

In light of these similarities, it remains elusive why KdpFABC includes an additional channel-like subunit in the transport process. The SF of KdpA is likely crucial to the particularly high substrate affinity and specificity. Due to its low sequence conservation to the classical TVGYG motif of tetrameric K$^+$ channels, SF discrimination in KdpA was suggested to base on a different mechanism[12,16]. Functional data already showed an absolute selectivity of KdpFABC for K$^+$, and, contrary to most K$^+$ channels, wild-type KdpFABC even excludes Rb$^+$. Specific mutations in the SF only allow the passage of Rb$^+$ and NH$_4^+$, and under no circumstances is Na$^+$ permitted[9,10]. A question not addressed here is how K$^+$ ions so selectively pass the SF to reach the intersubunit tunnel. Ion permeation in K$^+$ channels with the canonical TVGYG SFs is mediated by a Coulomb knock-on[31,32]. Despite the significantly altered selectivity filter, we observe the occupancy of three coordination sites (S1, S3, S4) in the SF, and similarly optimal coordination moieties for K$^+$ binding in the S2 site. While cryo-EM data do not allow to precisely quantify the occupancies of bound ligands, the presence of densities at three SF-binding sites, which are stably and simultaneously occupied in our MD simulations, strongly suggests a mechanism for ion entry into KdpFABC similar to that found in tetrameric K$^+$ channels. If so, this might explain the high selectivity of KdpFABC when compared to other SKT members, which showed fewer coordination sites[48,49]. Compared to canonical K$^+$ channels and the other SKT members, KdpFABC has a very low $K_m$ for K$^+$ transport of 2 μM[8]. The low $K_m$, essential for the function of KdpFABC as a high-affinity K$^+$ transporter, could be enabled by the energy minimum at the PBS in KdpB, which may augment the energetics of the SF knock-on to grant efficient ion progression. Furthermore, KdpC could promote high-affinity K$^+$ binding, as its sequence is related to β-subunits of other P-type ATPases, which are involved in substrate binding[7,50].

The data presented here verifies that ion transport in KdpFABC occurs via the unique intersubunit tunnel through KdpA and KdpB, and allows us to describe how ATP hydrolysis and K$^+$ transport are coupled in the chimeric complex. Moreover, we propose an intricate molecular system for the forwarding of ions into the CBS for transport, with key elements conserved in other P-type ATPases.

## Methods

**Cloning and protein production**. *Escherichia coli kdpFABC* was brought into expression vector pBXC3H using FX cloning[51]. Point mutations based on this construct were generated by site-directed mutagenesis, creating different variants listed in Table 1.

KdpFABC variants were produced in *E. coli* LB2003 cells (available from the Hänelt group upon request) transformed with plasmids encoding the relevant variant and grown in 6 or 12 l KML with 100 μg/ml ampicillin. Cultures were inoculated to $OD_{600}$ 0.1, induced with 0.002% L-arabinose at $OD_{600}$ 1.2, and harvested one hour after induction. For functional studies, KdpFABC variants were produced, purified, and characterized in parallel with wild-type KdpFABC to ensure comparability of measurements.

**KdpFABC purification.** Purification of KdpFABC variants for structural studies was performed as previously described for wild-type KdpFABC[17]. Purification of KdpFABC variants for functional studies was performed identically, omitting the final size exclusion chromatography step after anion exchange chromatography.

**Cryo-EM sample and grid preparation.** For the cryo-EM sample resulting in the 3.1 Å structure in the E1·ATP state with $K^+$, $KdpFAB_{D307N}C$ was used. The mutation $KdpB_{D307N}$ prevents ATP hydrolysis and stalls the complex in the E1 state[52]. Purified protein was concentrated to 3.1 mg/ml, and supplemented with 50 mM KCl and 5 mM AMPPCP to stabilize the complex in the E1·ATP conformation prior to grid preparation.

For the cryo-EM sample resulting in the 3.2 Å structure in the E1·ATP state with $Rb^+$, $KdpFA_{G232D}B_{S162A}C$ was used. The mutation $KdpA_{G232D}$ in the KdpA SF allows the passage of $Rb^{+9,10}$, while $KdpB_{S162}$ is the site of an inhibitory phosphorylation that inhibits turnover[16,21]. Purified protein was concentrated to 4 mg/ml, and supplemented with 100 mM RbCl and 1 mM ATP before incubation for 5 min at room temperature to displace any remaining $K^+$ in the complex with $Rb^+$. Turnover was stopped by the addition of 10 mM AMPPCP to stabilize the complex in the E1·ATP conformation prior to grid preparation.

Holey-carbon cryo-EM grids (Quantifoil Au R1.2/1.3, 300 mesh) were twice glow-discharged at 15 mA for 45 s. 2.8 μl of KdpFABC sample were applied to grids, blotted for 2–6 s in a VitroBot (Mark IV, ThermoFisher) at 4 °C and 100% humidity, and subsequently plunge-frozen in liquid ethane and stored in liquid nitrogen.

**Cryo-EM data collection.** Cryo-EM data were automatically collected using SerialEM software[53,54] (Thermo Fisher Scientific) on a 200 keV Talos Arctica microscope (Thermo Fisher Scientific) equipped with a post-column energy filter (Gatan) in zero-loss mode with a 20 eV slit and a 100 μM aperture with a K2 Summit detector (Gatan). Images were recorded at a pixel size of 1.012 Å (calibrated magnification of 49407×), a defocus range from −0.2 to −2.0 μm, an exposure time of 9 s, a subframe exposure time of 150 ms (60 frames), and a total electron exposure on the specimen level of about 52 electrons per $Å^2$. Data collection was optimized by restricting acquisition to regions displaying optimal sample thickness using an in-house written script[55], and the data quality monitored by on-the-fly processing using FOCUS software[56].

**Cryo-EM data processing.** The SBGrid[57] software package tool was used to manage the software packages.

$K^+$-*loaded* $KdpFAB_{D307N}C$. A total of 5,831 dose-fractioned cryo-EM images were recorded and subjected to motion-correction and dose-weighting of frames by MotionCor2[58]. The CTF parameters were estimated on the movie frames by ctffind4.1.13[59]. Bad images showing contamination, a defocus below −0.5 or above −2.0 μm, or a bad CTF estimation were discarded, resulting in 3,530 images used for further analysis with the software package RELION 3.0.8[60]. First, crYOLO 1.3.1[61] was used to automatically pick 331,673 particles using a loose threshold. Particle coordinates were imported in RELION 3.0.8[60], and the particles were extracted with a box size of 240 pixels. False positives or particles belonging to low-abundance classes were removed in several rounds of 2D classification, resulting in 249,092 particles. For 3D classification and refinement, the map of the previously generated E1 conformation EMD-0257[17] was used as reference for the first round, and the best output class was used in subsequent jobs in an iterative way. Particles belonging to the best classes were selected, resulting in 160,776 particles. Sequentially, several rounds of CTF refinement[60] were performed, using per-particle CTF estimation. In the last refinement iteration, a mask excluding the micelle was used and the refinement was continued until convergence (focused refinement), yielding a final map for the E1·ATP state at a resolution of 3.5 Å, and 3.1 Å after post-processing and masking, sharpened using an isotropic b-factor of −92 $Å^2$. No symmetry was imposed during 3D classification or refinement.

$Rb^+$-*loaded* $KdpFA_{G232D}B_{S162A}C$. A total of 22,046 dose-fractioned cryo-EM images were recorded and subjected to motion-correction and dose-weighting of frames by MotionCor2[58]. The CTF parameters were estimated on the movie frames by ctffind4.1.13[59]. Bad images showing contamination, a defocus below −0.5 or above −2.0 μm or a bad CTF estimation were discarded, resulting in 14,947 images used for further analysis with the software package RELION 3.1.0[60]. First, crYOLO 1.3.1[61] was used to automatically pick 756,834 particles using a loose threshold. Particle coordinates were imported in RELION 3.1.0[60], and the particles were extracted with a box size of 240 pixels. False positives or particles belonging to low-abundance classes were removed in several rounds of 2D classification, resulting in

469,824 particles. For 3D classification and refinement, the map of the previously generated E1 conformation EMD-0257[17] was used as reference for the first round, and the best output class was used in subsequent jobs in an iterative way. Particles belonging to the best class were selected, resulting in 276,980 particles. Sequentially, several rounds of CTF refinement[60] were performed, using per-particle CTF estimation, before subjecting the dataset to a round of focused 3D classification with no image alignment, using a mask on the flexible AN domains of KdpB[62], which resulted in a cleaned dataset of 196,682 particles. This dataset was subjected to another round of CTF refinement. In the last refinement iteration, a mask excluding the micelle was used and the refinement was continued until convergence (focused refinement), yielding a final map at a resolution of 3.6 Å before masking and 3.2 Å after masking, sharpened using an isotropic b-factor of −130 $Å^2$. No symmetry was imposed during 3D classification or refinement.

For both datasets, local resolution estimates were calculated by RELION. All resolutions were estimated using the 0.143 cutoff criterion[63] with gold-standard Fourier shell correlation (FSC) between two independently refined half maps. During post-processing, the approach of high-resolution noise substitution was used to correct for convolution effects of real-space masking on the FSC curve[64].

**Model building and validation.** The cryo-EM structure of KdpFABC in an E1 conformation 6HRA[17] was split into the membrane domain: KdpA, KdpB (residues 1–88, 216–274, and 570–682,), KdpC, and KdpF, and the three cytosolic domains of KdpB: KdpB-P (residues 275–314 and 451–569), KdpB-N (residues 315–450) and KdpB-A (residues 89–215). Initially, all fragments were docked into the obtained cryo-EM map using UCSF Chimera[65]. The connections between the four KdpB segments were modeled manually in Coot[66]. The initial model was then subjected to an iterative process of real-space refinement using Phenix.real_space_refinement with secondary structure restraints[67,68], followed by manual inspection and adjustments in Coot[66]. Rubidium ions, cardiolipins, and AMPPCP were added into the cryo-EM maps. The final model was refined in real space with Phenix.real_space_refinement with secondary structure restraints[67,68]. For validation of the refinement, FSC (FSC$_{sum}$) between the refined model and the final map was determined. To monitor the effects of potential over-fitting, random shifts (up to 0.5 Å) were introduced into the coordinates of the final model, followed by refinement against the first unfiltered half-map. The FSC between this shaken-refined model and the first half-map used during validation refinement is termed FSC$_{work}$, and the FSC against the second half-map, which was not used at any point during refinement, is termed FSC$_{free}$. The marginal gap between the curves describing FSC$_{work}$ and FSC$_{free}$ indicate no over-fitting of the model. The geometries of the atomic model was evaluated using MolProbity[69].

**Tunnel calculations.** Pore calculations through the selectivity filter of the channel-like subunit KdpA were performed using the software HOLE[70]. For calculations, the PDB of KdpA, without cofactors or other subunits and aligned with the translocation pore parallel to the z-axis, was probed using a Conolly probe of radius 0.9 Å.

Radius calculations of the intersubunit tunnel were performed using CAVER Analyst[71]. Calculations were performed on PDB models without ions using a probe radius of 0.9 Å to allow estimation of constrictions in the tunnel. These settings result in a set of unspecific tunnels, from which the relevant tunnel was selected via comparison with previously calculated tunnels in 6HRA[17]. The intersubunit tunnel between KdpA and KdpB was visualized using HOLLOW with a probe radius of 1.1 Å and a starting point immediately below the KdpA SF[72].

**Ion density analysis.** Identified densities in the intersubunit tunnel of the $K^+$-loaded and the $Rb^+$-loaded E1·ATP map presented here were low-pass filtered to a resolution of 3.2 Å. For these filtered maps, the ion densities in the CBS, the intersubunit tunnel, and SF were quantified in Coot[66], by determining the minimum sigma for these densities to be visualized. For global comparison of all possible ion densities, these values were then normalized against the value obtained for the S3 position in the $K^+$-loaded map for each applied filter. For comparisons of individual densities, values were normalized to the value obtained for the specific ion in the $K^+$-loaded map.

The expected scattering ratio of Rb/K was estimated as follow. A simplified estimation is obtained by using the formula $Z^4/3$, which is independent of the electron voltage[24–26]. With Z = 19 for K and 37 for Rb, the Rb/K ratio is 2.43. A better approximation, valid for elements with Z < ~100 and taking into account the relativistic speed of the electrons, which makes the ratio dependent on the acceleration voltage, is described in Reimer and Kohl[25], formula 5.41. Using beta = v/c = 0.7 for 200 kV gives a Rb/K ratio of 2.59 (v is the speed of the electrons, c is the speed of light). Based on these calculations, using an expected factor of 2.5 seems appropriate.

**ATPase assay.** ATP hydrolysis by purified KdpFABC variants was observed by malachite green ATPase assay[73]. In brief, 2 mM ATP, 1 mM KCl, and indicated lipids or inhibitors were mixed to the respective conditions. Reactions were started by the addition of 0.5–1.0 μg protein and carried out for 5 min at 37 °C. Lipid stocks were prepared as previously described[36].

**Growth complementation assay.** The growth complementation assays were performed as previously described[74]. In brief, *E. coli* LB2003, a strain lacking all endogenous K+ uptake systems, was transformed with plasmids encoding His-tagged KdpFABC variants. LB2003 transformed with empty vector pBAD18 or plasmid pBXC3H-KdpFAB$_{D307N}$C, encoding for an inactive variant, served as negative controls. Growth was monitored for 24 h at different K+ concentrations (1–115 mM, referred to as K1–K115). At K10 and below, the strain only grows sufficiently if the produced protein complements the lacking K+ transport systems. Protein production was confirmed by SDS-PAGE and subsequent western blotting analysis of a K30 sample after 24 h using an anti-His antibody from mouse (dilution 1:3000, Sigma–Aldrich, cat.no. H1029) and secondary anti-mouse IgG-HRP antibody produced in goat (dilution 1:20,000, Sigma–Aldrich, cat.no. A2554).

**Reconstitution and ACMA-based liposome transport assay.** Liposomes were prepared from *E. coli* polar lipid extract. Five milligrams of dry lipids were resuspended to 10 mg/ml in hot (55 °C) reconstitution buffer (15 mM Tris-HCl pH 7.5, 2 mM MgCl$_2$, 50 mM KCl), homogenized by sonication and subjected to three freeze-thaw cycles. Liposomes were diluted to a concentration of 4 mg/ml in reconstitution buffer, and extruded to a diameter of 400 nm. Extruded liposomes were destabilized by titration of 10% Triton-X-100 to saturation. Detergent-solubilized protein was added to destabilized liposomes at a mass ratio of 1:5 and incubated at room temperature for 30 min. Detergent was removed by the gradual addition of Bio-Beads™ SM-2 in 40 mg/ml wet weight intervals, with incubation times between additions as follows: 15 min at room temperature, 15 min at 4 °C, 30 min at 4 °C, 1 h at 4 °C, and overnight at 4 °C. In the final detergent removal step, 60 mg/ml Bio-Beads were added and incubated for 1 h at 4 °C. The supernatant was extracted, proteoliposomes pelleted by ultracentrifugation for 30 min at 80,000 × *g*, washed once in reconstitution buffer and pelleted again by ultra-centrifugation for 30 min at 80,000 × *g*. Proteoliposomes were resuspended to a final concentration of 5 mg/ml in reconstitution buffer for further experiments.

K+ translocation by reconstituted KdpFABC was tested by a modified ACMA-based flux assay[75]. Proteoliposomes were diluted to 1 mg/ml and supplemented with 2 μM ACMA (Ex. 410 nm, Em. 480 nm) and 1 μM CCCP. Transport was initialized by the addition of 0.2 mM ATP, and ended by the addition of 0.2 μM valinomycin. Transport rates were obtained by linear fitting of the initial 50 s of transport.

**Molecular dynamics simulations**
*CGMD simulations.* CG systems were built using the deposited coordinates for PDBs 5MRW, 6HRA, 6HRB, 7BGY, and 7NNL. Protein atoms were converted to the CG Martini force field[76,77], using the Martini 3.0.b.3.2 open beta[78]. Additional bonds of 500 kJ mol$^{-1}$ nm$^{-2}$ were applied between all protein backbone beads within 1 nm. Proteins were built into membranes composed of 10% CL, 23% POPG, and 67% POPE, using the insane protocol[79]. CL parameters were used as described in Corey et al., 2021[28]. All systems were solvated with Martini waters and Na+ and Cl− ions to a neutral charge and 150 mM. Systems were minimized using the steepest descents method, followed by 1 ns equilibration with 5 fs time steps, then by 100 ns equilibration with 20 fs time steps, before 5 × 4 μs production simulations using 20 fs time steps, all in the NPT ensemble with the V-rescale thermostat and semi-isotropic Parrinello-Rahman pressure coupling[80,81].

Protein-lipid contact analyses were run on the final 3 μs of each trajectory, where contact was set at an inter-residue minimum distance of 0.6 nm, and the fraction of frames for which contact occurs plotted.

*Atomistic MD simulations.* For simulations of KdpFABC, the structural model of 7NNL was truncated by removing the KdpB soluble domains in the regions KdpB$_{V100-G196}$ and KdpB$_{D302-L558}$. Positional restraints (1000 kJ mol$^{-1}$ nm$^{-2}$) were applied to backbone atoms C, C$_\alpha$, and N to prevent shifts in the input structure. Alternatively, simulations were built using the full-length KdpFABC in the E2 state 7BGY. Protein atoms were described using the CHARMM36 force field, and inserted into a 67% POPE, 23% POPG, 10% CL lipid bilayer solvated with TIP3P water and 150 mM K+/Cl− using CHARMM-GUI[82–84]. For the majority of the simulations, the protonation states of all side chains were set to default apart from KdpB$_{K586}$, and KdpA$_{E370}$, which had pKas of 7.5, and 9.1, respectively, based on analysis with PropKa3.1[85]. Additional systems were built in which KdpB$_{D583}$ and KdpB$_{K586}$ were either both charged or both neutral, as well as a neutral KdpB$_{E370}$ (Supplementary Fig. 14a, b).

Systems were energy minimized using the steepest descents method, and subsequently equilibrated with positional restraints on heavy atoms for 100 ps in the NPT ensemble at 310 K with the V-rescale thermostat and semi-isotropic Parrinello-Rahman pressure coupling[80,81]. For both stages, positional restraints of 1000 kJ mol$^{-1}$ nm$^{-2}$ were applied to the K+ in the tunnel. Production simulations were run in triplicate with 2 fs time steps for *ca.* 250 ns unless specified.

Ion density analyses were run using the VolMap tool of VMD[86] with the default settings, taken over 3 × 250 ns of simulation for each system.

All atomistic and CGMD simulations were run in Gromacs 2019[87].

**APBS calculations.** Adaptive Poisson-Boltzmann Solver (APBS) calculations[88] were performed for K+ along the intersubunit tunnel. The first set of calculations

(Fig. 3b) were for K+ positions taken from a 20 ns MD relaxation simulation. Note that positions from multiple MD snapshots were taken to provide maximum coverage of the tunnel (see Supplementary Fig. 10b for positions). Alternatively, the input structural protein and ion coordinates from 7NNL were used, following steepest descents energy minimization (Supplementary Fig. 10c).

For each K+ position, a system was built-in CHARMM36 comprising KdpFABC and the single K+ ion at that position. The coordinates were then relaxed using steepest descents energy minimization, and processed using PDB2PQR[89], setting KdpA$_{K586}$ and KdpB$_{E370}$ to neutral states. APBS values for K+ were determined, assuming an ionic radius of 2.172 Å. The default settings for APBS were used as produced by the APBS webserver (https://server.poissonboltzmann.org/).

**Sequence alignment.** Conservation of sequences in KdpB across different species was evaluated by sequence comparison using Clustal Omega[90], with sequences from previously published comparisons[16] to ensure species diversity.

**Figure preparation.** Figures were prepared using UCSF Chimera[65], UCSF ChimeraX[91], PyMOL, VMD[86], OriginPro 2016, and GraphPad Prism 8.

**Data availability**
The cryo-EM map and model of K+-loaded KdpFAB$_{D307N}$C E1·ATP were deposited in the EMDB and wwPDB with accession codes EMD-12478 and 7NNL, respectively. The cryo-EM map and model of Rb+-loaded KdpFA$_{G232D}$B$_{S162A}$C E1·ATP were deposited in the EMDB and wwPDB with accession codes EMD-12482 and 7NNP, respectively. Source data are provided with this paper for all functional data; the minimum dataset is provided in a source data file. Further data supporting the findings in this manuscript are available upon reasonable request. Source data are provided with this paper.

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

## Acknowledgements
We thank Werner Kühlbrandt for access to the vitrobot and Sonja Welsch and Janina Stautz for freezing the cryo-EM grids. J.M.S. thanks Paul JN Böhm for assistance with cloning, cell growth, and purification. C.P. thanks Jan Rheinberger for EM support and Michiel Punter for IT support. P.J.S. acknowledges the University of Warwick Scientific Computing Research Technology Platform for computational access. The work was funded by the NWO Veni grant (722.017.001) and the NWO Start-Up grant (740.018.016) to C.P. as well as by the DFG Emmy Noether grant (HA6322/3-1), and the Life Science Bridge Award by the Aventis Foundation to IH. Research in P.J.S.'s lab is funded by Wellcome (208361/Z/17/Z), the MRC (MR/S009213/1) and BBSRC (BB/P01948X/1, BB/R002517/1, and BB/S003339/1). J.M.S. is funded by the State of Hesse in the LOEWE Schwerpunkt TRABITA and R.A.C. is funded by Wellcome (208361/Z/17/Z). This project made use of time on ARCHER, ARCHER2, and JADE granted via the UK High-End Computing Consortium for Biomolecular Simulation, HECBioSim (http://hecbiosim.ac.uk), supported by EPSRC (grant no. EP/R029407/1).

## Author contributions
J.M.S.: Validation, formal analysis, investigation, writing—original draft, writing—review and editing, visualization. R.A.C.: Validation, formal analysis, investigation, writing—original draft, writing—review and editing, visualization. L.H.: Validation, formal analysis, investigation, writing—original draft, writing—review and editing, visualization. C.S.: Validation, formal analysis, investigation, writing—original draft, writing—review and editing, visualization. P.J.S.: Conceptualization, writing—review and editing, supervision, project administration, funding acquisition. C.P.: Conceptualization, writing—review and editing, supervision, project administration, funding acquisition. I.H.: Conceptualization, writing—review and editing, supervision, project administration, funding acquisition.

## Funding

## Competing interests
The authors declare no competing interests.
