## [Peer Review File · Nature Communications]

REVIEWER COMMENTS

Reviewer #1 (Remarks to the Author):

Silberberg and colleagues present structural and functional analyses of the KdpFABC K⁺ transporter. Following upon earlier work from the same groups, the authors suggest that the K⁺ transport pathway traverses through an intersubunit tunnel. Within this tunnel, several non-protein densities in cryo-EM structures determined in the presence of K⁺ or Rb⁺ are assigned as ions based on structural analyses and molecular dynamics simulations. Within the tunnel, a conserved phenylalanine is identified that is critical for coupling ATPase activity to K⁺ transport. Together, these results provide an improved understanding of the molecular mechanisms underlying the function of this interesting transporter. However, I have several criticisms regarding technical aspects of the study that should be addressed prior to publication.

Comments:

1. The authors determine cryo-EM structures of KdpFABC in the presence of K⁺ and Rb⁺ with the goal of identifying ion binding sites in the protein translocation pathway. The authors state that Rb⁺ scatters electrons more strongly than K⁺ due to its larger atomic number. What is the expected difference in scattering between the ions for a 200 KeV electron microscope? Was a difference map calculated that would reveal peaks corresponding to the more strongly scattered Rb⁺ ions?
2. Following from point 1, the presentation of the increased ion density presented in Figure 2d is misleading. The authors should replace Figure 2d with Figure S8d, which compares the relative strength of the density peaks at all of the putative ion-binding sites. Figure 8d demonstrates that the benefit of Rb⁺ in terms of peak intensity is more variable and in most cases much more modest. The authors discuss this point in the text, but the main text figure should more clearly present it.
3. The error bars presented in Figures 2d and S8d are not meaningful. If the authors wish to have error bars, density maps should be calculated using different reconstruction algorithms (i.e. relion, cryosparc and cistem).
4. Figure 1d detracts from the presentation of the analysis of the ion translocation pathway presented in figures 1a-c. The role of cardiolipin in KdpFABC activity is a nice result and should be presented as a separate figure incorporating some of the results presented in ED Figure 1.
5. How are the ion occupancies calculated for the simulation data in Figure 3a? If these represent stable binding sites in the simulations, then a thresholded density profile for the ions should be presented (as in Figure 4d).
6. How do authors determine that all three positions in the selectivity filter are simultaneously occupied? Calculating ligand occupancy from Cryo-EM density maps is very difficult.
7. The combination of utilizing cryo-EM structures determined with different ions along with MD simulations provides two complementary approaches to assigning ion-binding sites in cryo-EM structures. As the authors have clearly thought about this issue, it would be helpful if the authors would comment on the difficulties in interpreting non-protein densities in cryo-EM structures and approaches that they and other groups have used to better distinguish ion densities from ordered water molecules and noise.

Minor comments:

1. The shaded spheres of the EM data are too similar to the filled spheres of the MD data in Figure 3A makes it very hard to distinguish the position of the ion-binding sites in derived from the two techniques.
2. Distances between the putative ion-binding sites and the coordinating residues should be included in Figure S7. Especially for residues such as R493 which would not be predicted to make favorable interactions with the ions.

Reviewer #2 (Remarks to the Author):

This is an excellent manuscript providing valuable new information on the structure and mechanism of the KdpFABC-ATPase in particular, and P-type ATPases in general. The authors have made clever use of different substrates to lock the enzyme into particular conformational states or to locate the positions of substrate binding sites and ion pathways. By employing Rb⁺ ions as a K⁺ congener, and making use of the fact that Rb⁺ scatters electrons more strongly than K⁺, they were able to unambiguously attribute the electron density along the intersubunit tunnel to the substrate ions. By using the nonhydrolyzable ATP analogue AMPPCP they were able to trap the enzyme in phosphorylated intermediate states. To complement their experimental work, they also performed molecular dynamics simulations. These showed good agreement with the structures they determined by cryo-electron microscopy, particularly in regard to the positions of the substrate ions. I have a few points for consideration by the authors.

p. 3, line 63

"conserved TGES motif"

Can the authors provide a reference showing that this motif is conserved? Where is it conserved, e.g. across all bacterial KdpB pumps or across all P-type ATPases? A possible reference could be V. Dubey et al., J. Mol. Biol. 433 (2021) 167008.

p. 3, lines 60-63

"A unique feature of KdpB among P-type ATPases is that it is subject to an inhibitory serine phosphorylation in the A domain...which lies in the conserved TGES motif, responsible for dephosphorylation of the E2-P state in the catalytic cycle."

However, the TGES motif is also present other pumps, including the Na⁺,K⁺-ATPase (M. Sweet et al., eLife 9 (2020) e55480.) The presence of a TGES motif itself is not unique to KdpB-ATPases. Do the authors mean that serine phosphorylation of the TGES motif is unique to KdpB? If so, can they suggest a reason why this motif doesn't undergo serine phosphorylation in the Na⁺,K⁺-ATPase or other P-type ATPases? The E2-P state must undergo dephosphorylation in the catalytic cycle of all P-type ATPases.

p. 3, lines 76-77

"Initially, the coupling helix model was proposed,..."

Who proposed it? A reference is required here.

p. 3, lines 77-78

"the presence of K⁺ ions in the SF of KdpA is connected to KdpB via a Grotthus mechanism..."

A Grotthus mechanism is also referred to as a "proton wire" mechanism (e.g. T. E. De Coursey and V. V. Cherny, J. Gen. Physiol. 109 (1997) 415-434), where protons hop between water molecules. Is this what the authors mean here, or do they mean that K⁺ ions are hopping?

p. 5, line 120

Replace "stronger" with "strongly", i.e. it should be an adverb, not an adjective.

p. 6, line 131

Insert "to" between "prior" and "cryo-EM".

p. 11, lines 256-260

The authors suggest that K⁺ ions push each other forward via polarisation of a phenylalanine n electron cloud or via charge-charge repulsion.

In the field of ion channels this is commonly referred to as a "knock-on" mechanism (cf. T. Sumikama and S. Oiki, J. Physiol. Sci. 69 (2019) 919-930). It would be worth including a reference to knock-on mechanisms for ion channels.

Reviewer #3 (Remarks to the Author):

The authors present a combined cryo-EM and molecular dynamics study on the ion transport pathway of the ion pump KdpFABC. Different mechanisms had previously been proposed about the different roles of the ATP hydrolysis/pump domains and the channel domains found in the complex. This novel and important contribution provides strong evidence for the hypothesis of an inter-subunit ion passage that provides a continuous ion pathway from the channel to the pump domains. In this proposed mechanism, the primary role of the channel domain is to act as a filter to make sure only potassium ions pass to the pump region. A constriction region is identified between the subunits in which a key phenylalanine is located. This residue is proposed to be influenced by ions through cation- π interactions. A tentative functional cycle is presented that includes a full ion translocation event, and which also involves a transient reprotonation of a Lys/Asp pair that is suggested to guide the ions. This is an interesting and important study that merits publication, after the following concerns have been addressed.

First, it is intriguing that the AG232D mutant is required to facilitate Rb⁺ permeation. Canonical potassium channels (that have a structurally similar selectivity filter) typically permeate Rb⁺ ions without modifications. It would thus be of interest to discuss why this mutant is required in the case of KdpFABC. Also, would the mutation be expected to affect the selectivity for other ions such as Na⁺ as well?

Second, concerning Fig. 3b, the reported energy values require some critical discussion. If taken at face value, the difference of hundreds of kJ/mol along the permeation pathway would not be compatible with ion transport. Therefore, it remains somewhat unclear if the data as they are can be used to infer the overall affinity of the tunnel to ions, as is done in the current manuscript.

Finally, in the presentation of the simulation results, it remains unclear which results are obtained by coarse grained, and which by atomistic simulations. I would guess that the cardiolipin results were obtained from coarse-grained simulations and the remaining results from atomistic simulations, but this should be explicitly stated in the manuscript.

Reviewer #1 (Remarks to the Author):

Silberberg and colleagues present structural and functional analyses of the KdpFABC K⁺ transporter. Following upon earlier work from the same groups, the authors suggest that the K⁺ transport pathway traverses through an intersubunit tunnel. Within this tunnel, several non-protein densities in cryo-EM structures determined in the presence of K⁺ or Rb⁺ are assigned as ions based on structural analyses and molecular dynamics simulations. Within the tunnel, a conserved phenylalanine is identified that is critical for coupling ATPase activity to K⁺ transport. Together, these results provide an improved understanding of the molecular mechanisms underlying the function of this interesting transporter. However, I have several criticisms regarding technical aspects of the study that should be addressed prior to publication.

We thank the reviewer for their comments and suggestions, and have implemented the following adjustments in response to their points:

Comments:

1. The authors determine cryo-EM structures of KdpFABC in the presence of K⁺ and Rb⁺ with the goal of identifying ion binding sites in the protein translocation pathway. The authors state that Rb⁺ scatters electrons more strongly than K⁺ due to its larger atomic number. What is the expected difference in scattering between the ions for a 200 KeV electron microscope? Was a difference map calculated that would reveal peaks corresponding to the more strongly scattered Rb⁺ ions?

We thank the reviewer for their suggestions on this topic. As an explanation for the expected difference in scattering we have added the following paragraph in the methods section (lines 764-772): “A simplified estimation is obtained by using the formula $Z^4/3$, which is independent of the electron voltage (Franken et al., Small (2020), Egerton, *physica status solidi* (1979), Reimer and Kohl Book “Transmission Electron Microscopy”, 2008, formula 5.40). With $Z= 19$ for K and 37 for Rb, the Rb/K ratio is 2.43. A better approximation, valid for elements with $Z < 100$ and taking into account the relativistic speed of the electrons, which makes the ratio dependent on the acceleration voltage, is described in Reimer and Kohl, 2008, formula 5.41. Using $\beta=v/c=0.7$ for 200 kV gives a Rb/K ratio of 2.59. (v is the speed of the electrons, c is the speed of light). Based on these calculations, using an expected factor of 2.5 seems appropriate.” Further, we have added the expected factor for electron scattering of 2.5 when introducing the method (line 130) and in the description of the results (lines 273-274), along with the three references listed above. Taking into account differences in local resolution, occupancy, efficiency of substitution, etc. between the averaged data sets, the two-fold increase we observe is within the expected range of the theoretical value.

We have also calculated a difference map between the K⁺-loaded and Rb⁺-loaded maps, which reveals the described non-protein densities in SF S3 and S4, the CBS, as well as the SF mutation G232D (see screenshot below). We have opted not to include this in the manuscript, as we think the quantification carries more information.

Difference map – K⁺/Rb⁺ loading (pink) with assigned Rb⁺ ions in blue.

2. Following from point 1, the presentation of the increased ion density presented in Figure 2d is misleading. The authors should replace Figure 2d with Figure S8d, which compares the relative strength of the density

peaks at all of the putative ion-binding sites. Figure 8d demonstrates that the benefit of Rb⁺ in terms of peak intensity is more variable and in most cases much more modest. The authors discuss this point in the text, but the main text figure should more clearly present it.

While we agree with the reviewer that Figure S8d (now S9d) shows a more direct view of the effect Rb⁺ incorporation, we believe that by normalizing the ion intensities at the entrance of the tunnel and at the CBS, we show that the relative effect of Rb⁺ incorporation at both sites is the same and in accordance with the expected increase in scattering signal of about 2-2.5 fold. We think that this is the important takeaway from this experiment, which is why we opted to show the normalized data in the main figure (now Fig 3d), and discuss the less evident differences in the middle of the tunnel in the text. To increase the visibility of the non-normalized data, we have added a reference to S8d (now S9d) in the figure caption (line 253). We have added a sentence to our discussion of the more ambiguous densities (lines 282-283) to note that unambiguous ion assignment would require anomalous dispersion data, which provides more context to our analysis.

3. The error bars presented in Figures 2d and S8d are not meaningful. If the authors wish to have error bars, density maps should be calculated using different reconstruction algorithms (i.e. relion, cryosparc and cistem).

We thank the reviewer for this insight – we have tried to generate data using different reconstruction algorithms. While we have found that cryoSPARC delivers very comparable results (see below), the map obtained with cisTEM was overall more poorly refined with the non-protein densities generally less well resolved. Here, despite the ions only becoming visible at considerably lower sigma, the effect of stronger Rb scattering in the SF and CBS is still visible. Rather than show results from only two reconstructions, we have therefore opted to show only the value comparing the two Relion maps low-pass filtered to a resolution of 3.2 Å without error bars. We consider this the most accurate comparison of the two density maps.

4. Figure 1d detracts from the presentation of the analysis of the ion translocation pathway presented in figures 1a-c. The role of cardiolipin in KdpFABC activity is a nice result and should be presented as a separate figure incorporating some of the results presented in ED Figure 1.

We are happy to see the reviewer's enthusiasm for this part of the manuscript. We have shifted panels from Fig 1 and ED Fig 1 to make a new Fig 2, showing CL in the structure, the MD results, the effect of the CL1 binding knockout, and the ATPase stimulation. The remaining panels have been consolidated into a supplementary figure (now S7). To describe the CL coordination we have added an additional sentence in the main text (lines 176-179).

5. How are the ion occupancies calculated for the simulation data in Figure 3a? If these represent stable binding sites in the simulations, then a thresholded density profile for the ions should be presented (as in Figure 4d).

The ion positions shown in Figure 3a (now Fig. 4a) are the final positions following a 20 ns MD relaxation of cryo-EM coordinates, as described in the section "APBS calculations" of the methods section. Thus, these

positions are absolute values, and not the product of occupancies over the course of a simulation such as in Figure 4d (now Fig 5d). We have added this distinction in the main text (line 286) and the caption to Figure 3 (now Fig. 4).

6. How do authors determine that all three positions in the selectivity filter are simultaneously occupied? Calculating ligand occupancy from Cryo-EM density maps is very difficult.

We thank the reviewer for this observation. We have responded to this by removing the phrasing “simultaneous occupancy” from the discussion, as this is indeed something we cannot determine from our cryo-EM density maps. However, we note that the simultaneous occupancy of these three sites is stable in our MD simulations, which lends credence to the idea of direct knock-on through the KdpA SF. Moreover, the identification of three binding sites in the SF, with a highly likely fourth site in S2, also indirectly supports such a knock-on mechanism similar to other K⁺ channels. As it stands, our data can however only hint at possible transduction mechanisms through the SF, and all discussed points remain speculative.

We have added to the SF discussion as follows (lines 590-599):

“While cryo-EM data does not allow to precisely quantify the occupancies of bound ligands, the presence of densities at three SF binding sites, which are stably and simultaneously occupied in our MD simulations, strongly suggests a mechanism for ion entry into KdpFABC similar to that found in tetrameric K⁺ channels.”

7. The combination of utilizing cryo-EM structures determined with different ions along with MD simulations provides two complementary approaches to assigning ion-binding sites in cryo-EM structures. As the authors have clearly thought about this issue, it would be helpful if the authors would comment on the difficulties in interpreting non-protein densities in cryo-EM structures and approaches that they and other groups have used to better distinguish ion densities from ordered water molecules and noise.

We thank the reviewer for this comment. While both methods used in this study have their strengths, and in combination paint a clear picture of ion transport through the intersubunit tunnel, they also have their limitations, which we also present in the manuscript. The use of Rb⁺ only shows a clear effect in sites with a strong ion coordination/occupancy. Meanwhile, the MD simulations show that almost all positions assigned are viable as ions, but are of course biased by the initial assignment of ions in the cryo-EM structure. In fact our simulations show that two of the positions identified from cryo-EM are not stable, suggesting that they might not represent ions.

To unequivocally identify individual densities as ions would require anomalous dispersion data from X-ray crystallography, for which Rb⁺ substitution would also be well suited, as its absorption edge is more easily reached than that of K⁺. We have added a sentence to reference this when discussing the assignment of ion densities in the cryo-EM map (lines 282-283).

Minor comments:

1. The shaded spheres of the EM data are too similar to the filled spheres of the MD data in Figure 3A makes it very hard to distinguish the position of the ion-binding sites in derived from the two techniques.

We have changed the color of the EM spheres to gray in this and all MD/cryo-EM comparison figures.

2. Distances between the putative ion-binding sites and the coordinating residues should be included in Figure S7. Especially for residues such as R493 which would not be predicted to make favorable interactions with the ions.

We agree with the reviewer that the distances to coordinating residues are important. Because adding them to the relevant figures would overload them, we have opted to add a supplementary table, in which the coordinating residues for all ions and the corresponding distances are listed (Supplementary Table 2).

R493 has a special role, as it is predicted to deflect ions entering through the KdpA SF into the intersubunit tunnel by electrostatic repulsion, as described in Stock et al., Nat. Comms. (2018). Therefore, the close proximity to the entering K⁺ in SF S4 is essential to the translocation process through the intersubunit tunnel. We have clarified this in the caption of S8b and S9b, and explicitly stated that the dotted lines represent ion-protein interactions (binding and/or repulsion) that are important to the transport process.

Reviewer #2 (Remarks to the Author):

This is an excellent manuscript providing valuable new information on the structure and mechanism of the KdpFABC-ATPase in particular, and P-type ATPases in general. The authors have made clever use of different substrates to lock the enzyme into particular conformational states or to locate the positions of substrate binding sites and ion pathways. By employing Rb⁺ ions as a K⁺ congener, and making use of the fact that Rb⁺ scatters electrons more strongly than K⁺, they were able to unambiguously attribute the electron density along the intersubunit tunnel to the substrate ions. By using the nonhydrolyzable ATP analogue AMPPCP they were able to trap the enzyme in phosphorylated intermediate states. To complement their experimental work, they also performed molecular dynamics simulations. These showed good agreement with the structures they determined by cryo-electron microscopy, particularly in regard to the positions of the substrate ions. I have a few points for consideration by the authors.

We thank the reviewer for their comments and enthusiasm towards our work, and have addressed the suggestions with the following changes:

p. 3, line 63 “conserved TGES motif” Can the authors provide a reference showing that this motif is conserved? Where is it conserved, e.g. across all bacterial KdpB pumps or across all P-type ATPases? A possible reference could be V. Dubey et al., J. Mol. Biol. 433 (2021) 167008.

The TGES motif is widely conserved among cation P-type ATPases. We have added this phrasing (lines 61-62), and references describing its identical role in three of the best-studied P-type ATPases (SERCA: Anthonisen et al., J. Biol. Chem. (2006); Na⁺/K⁺ ATPase: Toustrup-Jensen & Vilsen, J. Biol. Chem. (2003); H⁺ ATPase: Serrano & Portillo, BBA – Bioenerg. (1990)).

p. 3, lines 60-63 “A unique feature of KdpB among P-type ATPases is that it is subject to an inhibitory serine phosphorylation in the A domain... which lies in the conserved TGES motif, responsible for dephosphorylation of the E2-P state in the catalytic cycle.” However, the TGES motif is also present other pumps, including the Na⁺,K⁺-ATPase (M. Sweet et al., eLife 9 (2020) e55480.) The presence of a TGES motif itself is not unique to KdpB-ATPases. Do the authors mean that serine phosphorylation of the TGES motif is unique to KdpB? If so, can they suggest a reason why this motif doesn’t undergo serine phosphorylation in the Na⁺,K⁺-ATPase or other P-type ATPases? The E2-P state must undergo dephosphorylation in the catalytic cycle of all P-type ATPases.

While the Na⁺/K⁺ ATPase is also regulated by an inhibitory phosphorylation, this is not in the TGES motif (nor in the A domain at all for that matter). To our knowledge, the phosphorylation of the TGES motif is unique to KdpB.

We agree that this phosphorylation/feedback inhibition is a fascinating topic – likely it reflects the very limited range of conditions under which uptake by KdpFABC is physiologically required, compared to e.g. Na⁺/K⁺ ATPase. However, we feel that the focus of this manuscript is the ion pump cycle, not the regulation of the pump, therefore we do not want to delve too deeply into this topic. Because much remains unclear, any further discussion of this in the introduction would be speculative. Nonetheless, we have rephrased/added to this section, which now reads as follows (lines 61-66):

“Dephosphorylation is mediated by a TGES motif in the A domain, which is widely conserved among cation P-type ATPases¹⁷⁻¹⁹. A unique feature of the TGES motif of KdpB is that it is subject to an inhibitory serine phosphorylation (Escherichia coli (Ec)KdpB_{S162}). This phosphorylation has been shown to prevent excessive K⁺ uptake when external K⁺ is high, indicating that it limits uptake by KdpFABC to conditions where it is physiologically necessary^{15,20}.”

p. 3, lines 76-77 “Initially, the coupling helix model was proposed,...” Who proposed it? A reference is required here.

We have added the appropriate reference to Huang et al, Nature (2017), where this mechanism was first proposed.

p. 3, lines 77-78 “the presence of K⁺ ions in the SF of KdpA is connected to KdpB via a Grotthus mechanism...”
A Grotthus mechanism is also referred to as a “proton wire” mechanism (e.g. T. E. De Coursey and V. V. Cherny, *J. Gen. Physiol.* 109 (1997) 415-434), where protons hop between water molecules. Is this what the authors mean here, or do they mean that K⁺ ions are hopping?

The model proposed in Huang et al., *Nature* (2017) is indeed based on a proton wire/Grotthus mechanism to transfer a charge to the CBS. For completeness we have added both terminologies.

p. 5, line 120 Replace “stronger” with “strongly”, i.e. it should be an adverb, not an adjective.

The correction has been implemented (line 130).

p. 6, line 131 Insert “to” between “prior” and “cryo-EM”.

The correction has been implemented (line 152).

p. 11, lines 256-260 The authors suggest that K⁺ ions push each other forward via polarisation of a phenylalanine π electron cloud or via charge-charge repulsion. In the field of ion channels this is commonly referred to as a “knock-on” mechanism (cf. T. Sumikama and S. Oiki, *J. Physiol. Sci.* 69 (2019) 919-930). It would be worth including a reference to knock-on mechanisms for ion channels.

We thank the reviewer for this suggestion. We have added a comparison to the proposed Coulomb knock-on in the SF of K⁺ channels and references to Kopec et al., *Nat. Chem.* (2018) and Öster et al., *Sci. Adv.* (2019), which describe the MD simulations at the basis of the direct Coulomb knock-on theory (lines 350-351).

Reviewer #3 (Remarks to the Author):

The authors present a combined cryo-EM and molecular dynamics study on the ion transport pathway of the ion pump KdpFABC. Different mechanisms had previously been proposed about the different roles of the ATP hydrolysis/pump domains and the channel domains found in the complex. This novel and important contribution provides strong evidence for the hypothesis of an inter-subunit ion passage that provides a continuous ion pathway from the channel to the pump domains. In this proposed mechanism, the primary role of the channel domain is to act as a filter to make sure only potassium ions pass to the pump region. A constriction region is identified between the subunits in which a key phenylalanine is located. This residue is proposed to be influenced by ions through cation- π interactions. A tentative functional cycle is presented that includes a full ion translocation event, and which also involves a transient reprotonation of a Lys/Asp pair that is suggested to guide the ions. This is an interesting and important study that merits publication, after the following concerns have been addressed.

We thank the reviewer for their kind comments and suggestions, and have addressed their concerns as follows:

First, it is intriguing that the AG232D mutant is required to facilitate Rb⁺ permeation. Canonical potassium channels (that have a structurally similar selectivity filter) typically permeate Rb⁺ ions without modifications. It would thus be of interest to discuss why this mutant is required in the case of KdpFABC. Also, would the mutation be expected to affect the selectivity for other ions such as Na⁺ as well?

We thank the reviewer for their interest in this topic. The mutation G232D (and others) in the KdpA selectivity filter are well characterized in Schrader et al, *Biophys. J.* (2000) and van der Laan et al, *J. Bacteriol.* (2002), including its ion selectivity. Here, the authors find only two mutations in the SF that broaden the selectivity – G232D, which permits Rb⁺, and G232S, which permits NH₄⁺. No mutations were identified that permitted Na⁺. As mentioned in the discussion, the SF of KdpA deviates significantly from the canonical TVGYG motif, which possibly causes the high selectivity for K⁺ compared to other channels. The exact geometrical differences between the KdpA SF and the SF of e.g. KcsA have already been shown in Huang et al, *Nature* (2017). We have expanded on the background information concerning the KdpA SF in the discussion to provide more context (lines 578-586):

“The SF of KdpA is likely crucial to the particularly high substrate affinity and specificity. Due to its low sequence conservation to the classical TVGYG motif of tetrameric K⁺ channels, SF discrimination in KdpA was suggested to base on a different mechanism^{11,15}. Functional data already showed an absolute selectivity of KdpFABC for K⁺, and, contrary to most K⁺ channels, wild-type KdpFABC even excludes Rb⁺. Specific mutations in the SF only allow the passage of Rb⁺ and NH₄⁺, and under no circumstances is Na⁺ permitted^{8,9}. A question not addressed here is how K⁺ ions so selectively pass the SF to reach the intersubunit tunnel.”

In Supplementary Figure 9, we show the structural basis for Rb⁺ permissivity of the KdpA_{G232D} SF. We have added a table with the coordination distances for all ions in each structure (Supplementary Table 2), which reflects the change in hydration radius probably responsible for the altered selectivity.

However, we refrain from addressing in more depth how the selectivity differences between other K⁺ channels and KdpA could be explained. This would probably require MD simulations showing ion progression through the SF. However, even after many such studies on K⁺ channels, this process is still the topic of lively debate. We hope the reviewer agrees that conclusive commentary on this question in KdpA would therefore require studies that exceed the scope of this work.

Second, concerning Fig. 3b, the reported energy values require some critical discussion. If taken at face value, the difference of hundreds of kJ/mol along the permeation pathway would not be compatible with ion transport. Therefore, it remains somewhat unclear if the data as they are can be used to infer the overall affinity of the tunnel to ions, as is done in the current manuscript.

We thank the reviewer for this comment. The energy values reported here are based on APBS calculations, which estimate electrostatic potentials rather than free energies (which are a lot more costly to calculate). The reported energies omit key contributions to actual K⁺ transport, such as ion desolvation, fluctuations in ionic atmosphere and the effect of dispersion interactions. Furthermore, here we have modelled each K⁺ in isolation, which does not reflect the ion-filled tunnel we observe. We note that our energies are similar to those previously reported for K⁺ channels, e.g. Tai, Haider, Grottesi and Sansom. 2009. EBJ.

Here, rather than attempting to accurately describe the energetics of ion transport, we are using the APBS calculations to indicate which binding sites in the tunnel are of particular significance to the transport process.

The identification of clear minima at the SF and the PBS is sufficient for this purpose, which we then probe with further MD simulations and functional experiments.
In response to this comment, we have adjusted the text discussing the APBS data to read as follows (lines 289-295):

“An analysis of the coordination energy for K^+ along the tunnel by Adaptive Poisson-Boltzmann Solver (APBS) analysis was used as an approximate framework to compare ion positions in the tunnel, and indicated a favorable energy profile along the entire length, with notable energy minima in the KdpA SF, where K^+ is fully dehydrated by the protein, and, interestingly, at the end of the intersubunit tunnel, just before the KdpB CBS (Figure 3 b, Supplementary Figure 10 b,c).”

Finally, in the presentation of the simulation results, it remains unclear which results are obtained by coarse grained, and which by atomistic simulations. I would guess that the cardiolipin results were obtained from coarse-grained simulations and the remaining results from atomistic simulations, but this should be explicitly stated in the manuscript.

Good point! Indeed, only the cardiolipin simulations were performed as CG-simulations, all other simulations were atomistic. We have addressed this by clarifying in each situation (main text, methods, and figure captions) whether the presented results are from atomistic or coarse-grained simulations.

REVIEWERS' COMMENTS

Reviewer #1 (Remarks to the Author):

The authors have addressed all of my concerns and the manuscript is now suitable for publication.

Reviewer #3 (Remarks to the Author):

The authors have satisfactorily addressed my concerns.